# Fast and interpretable consensus clustering via minipatch learning

**Luqin Gan**[1]*, **Genevera I. Allen**[2,3]

**1** Department of Statistics, Rice University, Houston, Texas, United States of America, **2** Departments of Electrical and Computer Engineering, Statistics, and Computer Science, Rice University, Houston, Texas, United States of America, **3** Duncan Neurological Research Institute, Baylor College of Medicine, Houston, Texas, United States of America

* luain.gan@rice.edu

## Abstract

Consensus clustering has been widely used in bioinformatics and other applications to improve the accuracy, stability and reliability of clustering results. This approach ensembles cluster co-occurrences from multiple clustering runs on subsampled observations. For application to large-scale bioinformatics data, such as to discover cell types from single-cell sequencing data, for example, consensus clustering has two significant drawbacks: (i) computational inefficiency due to repeatedly applying clustering algorithms, and (ii) lack of interpretability into the important features for differentiating clusters. In this paper, we address these two challenges by developing IMPACC: Interpretable MiniPatch Adaptive Consensus Clustering. Our approach adopts three major innovations. We ensemble cluster co-occurrences from tiny subsets of both observations and features, termed minipatches, thus dramatically reducing computation time. Additionally, we develop adaptive sampling schemes for observations, which result in both improved reliability and computational savings, as well as adaptive sampling schemes of features, which lead to interpretable solutions by quickly learning the most relevant features that differentiate clusters. We study our approach on synthetic data and a variety of real large-scale bioinformatics data sets; results show that our approach not only yields more accurate and interpretable cluster solutions, but it also substantially improves computational efficiency compared to standard consensus clustering approaches.

## Author summary

Clustering seeks to discover groups in big data with wide applications across scientific domains, especially in bioinformatics. However, for huge and sparse data sets common with genomic sequencing technologies, clustering methods can suffer from unreliable results, lack of interpretability in terms of feature importance, and heavy computational costs. To solve these challenges, we propose an extension of consensus clustering that leverages minipatch learning, an ensemble learning framework with learners trained on tiny subsets of observations and features. With adaptive sampling frameworks on both features and observations, our method is able to achieve higher clustering accuracy and

**Data Availability Statement:** IMPACC, including source code and a tutorial, is freely available at https://github.com/DataSlingers/IMPACC. All data sets used in empirical study are uploaded to Kaggle: https://www.kaggle.com/ganluqin/impacc-data.

**Funding:** This study received funding from the National Science Foundation(DMS-1554821, https://www.nsf.gov/) and National Institutes of Health / National Institute of General Medicine (1R01GM140468, https://www.nigms.nih.gov/) received by G.I.A. The funders had no role in study design, data collection and analysis, decision to publish, or preparation of the manuscript.

**Competing interests:** The authors have declared that no competing interests exist.

reliability, as well as simultaneously identify scientifically important features that distinguish the clusters. In addition, we offer major computational improvements, with dramatically faster speed than our competitors. Our method is general and widely applicable to data sets from any field, and especially can offer superior performance when dealing with complex sparse and high dimensional data found in bioinformatics.

This is a *PLOS Computational Biology* Methods paper.

## Introduction

Consensus clustering is a widely used unsupervised ensemble method in the domains of bioinformatics, pattern recognition, image processing, and network analysis, among others. This method often outperforms conventional clustering algorithms by ensembling cluster co-occurrences from multiple clustering runs on subsampled observations [1]. However, consensus clustering has many drawbacks when dealing with large data sets typical in bioinformatics. These include computational inefficiency due to repeated clustering of very large data on multiple subsamples, degraded clustering accuracy due to high sensitivity to irrelevant features, as well as lack of interpretability. Consider, for example, the task of discovering cell types from single-cell RNA sequencing data. This data often contains tens-of-thousands of cells and genes, making consensus clustering computationally prohibitive. Additionally, only a small number of genes are typically responsible for differentiating cell types; consensus clustering considers all features and provides no interpretation of which features or genes may be important. Inspired by these challenges for large-scale bioinformatics data, we propose a novel approach of consensus clustering that utilizes tiny subsamples or minipatches as well as adaptive sampling schemes to speed computation and learn important features.

## Related work

Several types of consensus functions in ensemble clustering have been proposed, including co-association based function [2–5], hyper-graph partitioning [6–8], relabeling and voting approach [9–11], mixture model [12–14], and mutual information [15–17]. Co-association based function, such as consensus clustering, is faster in convergence and is more applicable to large-scale bioinformatics data sets. Our approach is based on consensus clustering, whose concept is straightforward. In order to achieve evidence accumulation, a consensus matrix is constructed from pairwise cluster co-occurrence, ranging in [0,1]. It is later regarded as a similarity matrix of the observations to obtain the final clustering results [18]. Closely related to our work, numerous variants of consensus clustering with adaptive subsampling strategies on observations have been proposed. For instance, Duarte et al. [19] update the sampling weights of objects with their degrees of confidence, which are subtracted by clustering the consensus matrix; Parvin et al. [20] compute sampling weights by the uncertainty of object assignments based on consensus indices' distances to 0.5; and Topchy et al. [21] adaptively subsample objects according to the consistency of clustering assignments in previous iterations. Besides adaptive sampling, Ren et al. [22] overweight the observations with high confusion, and assign the one-shot weights to obtain final clustering results. However, the existing sampling schemes focus on observations only and do not take feature relevance into consideration. So these

methods show inferior performance in the application to sparse data sets, where only a small set of features can significantly influence cluster assignments. Many clustering methods and pipelines have been proposed that specifically focus on single-cell RNA-seq data [23–27]. A popular approach, SC3 [23], employs consensus clustering by applying dimension reduction to the subsampled data and then applying K-means. Satija et al. [25] integrate dropout imputation and dimension reduction with a graph-based clustering algorithm. Another widely used and simple approach is to conduct tSNE dimension reduction followed by K-Means clustering [28]. Many have discussed the computational challenge of clustering large-scale single-cell sequencing data [28] and have sought to address this via dimension reduction. But clustering based on dimension reduced data is no longer directly interpretable; that is, one cannot determine which genes are directly responsible for differentiating cell type clusters. The motivation of our approach is not only to propose a computationally fast approach, but also to develop a method that has built-in feature interpretability to discover differentially expressed genes. A series of clustering algorithms have been proposed to add insights on feature importance. Some clustering algorithms conduct sparse feature selection through regularization within clustering algorithms. For example, sparse K-Means (sparseKM), sparse hierarchical clustering (sparseHC) [29] and sparse convex clustering [30,31] facilitate feature selection by solving a lasso type optimization problem. However, this type of sparse clustering algorithm is often slow and highly sensitive to hyper-parameter choices; thus, they face maybe computational challenges for large data. Another class of methods ranks features by their influence on results. The resulting sensitivity to the changes of one feature can be measured by the difference in silhouette widths of clustering results [32], the difference in the entropy of consensus matrices [33], or consistency of graph spectrum [34]. However, feature ranking methods have to measure the importance of each feature separately, which leads to extremely high computational costs. Additionally, [35] propose a post-hoc feature selection method that solves an optimization problem to determine important features within the standard consensus clustering algorithm; however, this approach suffers from major computational hurdles for large data. Therefore, we are motivated to propose an extension of consensus clustering to greatly improve clustering accuracy, provide model interpretability, and simultaneously ease the computational burden, by incorporating innovative adaptive sampling schemes on both features and observations with minipatch learning.

## Contributions

In this paper, we propose a novel methodology as an extension of consensus clustering, which demonstrates major advantages in large-scale bioinformatics data sets. Specifically, we seek to improve computational efficiency, provide interpretability in terms of feature importance, and at the same time improve clustering accuracy. We achieve these goals by leveraging the idea of minipatch learning [36–38], which is an ensemble of learners trained on tiny subsamples of both observations and features. Compared to only subsampling observations in existing consensus clustering ensembles, our approach offers significant computational savings by learning from many tiny data sets. In addition, we develop novel adaptive sampling schemes for both observations and features to concentrate learning on observations with uncertain cluster assignments and on features that are most important for separating clusters. This provides inherent interpretations for consensus clustering and also further improves the computational efficiency of the learning process. We test our novel methods and compare them to existing approaches through extensive simulations and four large real-data case studies from bioinformatics and imaging. Our results show major computational gains with our run time on the

same order as that of hierarchical clustering, as well as improved clustering accuracy, feature selection performance, and interpretability.

## Methods

Let $X \in R^{N \times M}$ be the data matrix of interest, with $M$ features measured over $N$ observations. $x_i \in R^M$ is the $M$-dimensional feature vector observed for sample $i$. Our goal is to partition the observations into disjoint homogeneous clusters, which can reflect the underlying data structures and patterns. We propose to extend popular consensus clustering techniques [39] to be able to detect clusters more accurately and computationally efficiently, in high-dimensional noisy data common in bioinformatics [40,41]. We also seek ways to ensure our clusters are interpretable through feature selection. To this end, we propose a number of innovations and improvements to consensus clustering outlined in our Minipatch Consensus Clustering framework in Algorithm 1. Similar to consensus clustering, our approach repeatedly subsamples the data, applies clustering, and records the $N \times N$ co-clustering membership matrix, $V$. It then ensembles all the co-clustering membership information together into the $N \times N$ consensus matrix $S$. This consensus matrix takes values in [0,1] indicating the proportion of times two observations are clustered together; it can be regarded as a similarity matrix for the observations. A perfect consensus matrix includes only entries of 0 or 1, where observations are always assigned to the same clusters; values in between indicate the (un)reliability of cluster assignments for each observation. To obtain final cluster assignments, one can cluster the estimated consensus matrix, which typically yields more accurate clusters than applying the standard, non-ensembled clustering algorithms [1].

While the core of our approach is identical to that of consensus clustering, we offer three major methodological innovations in Steps 1 and 2 of Algorithm 1 that yield 112 remarkably faster, more accurate, and interpretable results. Our first innovation is building cluster ensembles based on (n = 25%N, m = 10%M) tiny subsets with default of both observations and features termed minipatches [37–39]. Note that existing consensus clustering approaches form ensembles by subsampling typically 80% of observations and all the features for each ensemble member [42]. For large-scale bioinformatics data where the number of observations and features could be in the tens-of-thousands, repeated clustering of this large data is a major computational burden. Instead, our approach, termed Minipatch Consensus Clustering (MPCC), subsamples a tiny fraction of both observations and features and hence has obvious computational advantages. The computational complexity of MPCC in Algorithm 1 is $O(mn^2T + N^2)$, where $T$ is the total number of minipatches. Since $m$ and $n$ are very small, the dominating term is the $N^2$ computations required to update the consensus matrix. This compares very favorably to existing consensus clustering approaches. If the default of 80% of observations are subsampled in each run, then the time complexity is $O(MN^2T)$, which can be very slow for both large $N$ and large $M$ datasets. On the other hand, our method is comparable in complexity to hierarchical clustering, which is also $O(N^2)$ [43], but is perhaps slower than K-Means, which is $O(N)$ [44]. The proof of the time complexity is in S1 Text.

While MPCC offers dramatic computational improvements over standard consensus clustering, one may ask whether the results will be as accurate. We investigate and address this question from the perspective of how tiny subsamples of observations and separately features affect clustering results. First, note that if a tiny fraction of observations is subsampled, then by chance, some of the clusters may not be represented; this is especially the case for large $K$ or for uneven cluster sizes. Existing consensus clustering approaches typically apply a clustering algorithm with fixed $K$ to each subsample, but this practice would prove detrimental to our approach. Instead, we propose to choose the number of clusters in each minipatch adaptively.

While there are many techniques in the literature to do so that could be employed with our method [18,45], we are motivated to choose the number of clusters very quickly with nearly no additional computation. Hence, we propose to exclusively use hierarchical clustering on each minipatch and to cut the tree at the $h$ quantile of the dendrogram height to determine the number of clusters and cluster membership. This approach is not only fast but also adaptive to the number of clusters present in the minipatch, and the results change smoothly with cuts at different heights. Our empirical results reveal that this approach performs well on minipatches, and we specifically investigate its utility, sensitivity, and tuning of $h$ in S1 Text; importantly, we find that setting $h$ = .95 to nearly universally yields the best results, and hence we suggest fixing this value. Additionally, we provide details on hyper-parameters, tuning, and stopping criteria in S1 Text. Besides, we also explored other alternatives to determine the number of clusters in a minipatch, including selecting the cluster number with the highest silhouette score and using the oracle number of clusters $K$. However, the alternatives yield worse performance either in terms of clustering accuracy or computational time. Further details are in S1 Text.

Next, one may ask how subsampling the features in minipatches affects clustering accuracy. Obviously, for high-dimensional data in which only a small number of features are relevant for differentiating clusters, subsampling minipatches containing the correct features would improve results. We address such possibilities in the next section. But if this is not the case, would clustering accuracy suffer? Since we apply hierarchical clustering, which takes distances as input, we seek to understand how far off our distance input can be when we employ sub-samples of features. We consider this theoretically in S1 Text and empirically in the subsequent section. Our analysis and results reveal that while smaller minipatches yield faster computations, there may be a 164 slight trade-off in terms of clustering accuracy. Our empirical results in S1 Text 165 suggest that such a trade-off is generally slight or negligible, so we can typically utilize 166 smaller minipatches.

**Algorithm 1:** Minipatch Consensus Clustering
**Input:** *X*, *n*, *m*, $V^{(0)}$, $D^{(0)}$, *h*; **while** *stopping criteria not meet* **do**
1. Obtain minipatch $X_{It,Ft}$ $\in$ R$^{n \times m}$ by subsampling *n* observations $I_t \subset$ {1,...,*N*} and *m* featrues $F_t \subset$ {1,...,*M*}, without replacement;
• *MPCC subsamples uniformly at random;*
• *MPACC uses the adaptive observation sampling scheme only;*
• *IMPACC uses both adaptive feature and observation sampling schemes simultaneously;*
2. Obtain estimated clustering result C$^{(t)}$ by fitting hierarchical clustering to $X_{It,Ft}$ and cut tree at *h* height quantile;
3. Update co-clustering membership matrix *V* and co-sampling matrix *D*:

$$V^{(t)}(i,i') = V^{(t-1)}(i,i') + I(C_i^{(t)} = C_{i'}^{(t)}); \ D^{(t)}(i,i') = D^{(t-1)}(i,i') + I(i \in I_t, i' \in I_t)$$

end
Calculate consensus matrix $S(i,i') = \frac{V^{(T)}(i,i')}{\max(1, D^{(T)}(i,i'))}$;
Obtain final clustering result $\hat{\Pi}$ by using *S* as a similarity matrix;
**Output:** *S*, $\hat{\Pi}$.

We have introduced minipatch consensus clustering (MPCC) using random subsamples of both features and observations. The advantage of this approach is its computational speed, which is on the order of standard clustering approaches such as hierarchical and spectral clustering, as suggested by our empirical results in Results section. But, one may ask whether clustering results can be improved by perhaps optimally sampling observations and/or features instead of random sampling. Some

have suggested such possibilities in the context of consensus clustering [19–22]; we explore it and develop new approaches for this in the following sections.

## Minipatch Adaptive Consensus Clustering (MPACC)

One may ask whether it is possible to improve upon minipatch consensus clustering in terms of both speed and clustering accuracy by adaptively sampling observations. For example, we may want to sample observations that are not well clustered more frequently to learn their cluster assignments faster. In the method MiniPatch Adaptive Consensus Clustering (MPACC), we propose to dynamically update sampling weights, with a focus on observations that are difficult to be clustered and that are less frequently sampled. In addition, we leverage the adaptive weights by designing a novel observation sampling scheme. Specifically, we propose to update observation weights by adjusted confusion values dynamically, with a default learning rate $\alpha_I = 0.5$. To measure the level of clustering uncertainty, confusion values are derived from consensus matrix, given by $\text{conufusion}_i = \frac{1}{N}\sum_{i'=1}^{N} S(i, i')(1 - S(i, i'))$ for observation $i$. A larger confusion value near 0.25 indicates poorer clustering with unstable assignments, and the minimum confusion value 0 suggests perfect clustering. Note that confusions tend to grow with iterations because more consensus values are updated from the initial value 0. Therefore, a large confusion value due to oversampling cannot truly reflect the level of uncertainty. To eliminate bias caused by oversampling and to upweight less frequently sampled observations, we further adjust confusion values by sampling frequencies of observations in previous iterations, as presented in Algorithm 2. The next question is, how do we leverage the weights to dynamically construct minipatches as the number of iterations grows? A straightforward solution is to probabilistically subsample with probability (*Prob*) proportional to the weights. But the problem with this approach is that the clustering performance will be compromised if we only tend to sample uncertain and difficult observations. To resolve such drawback, we develop an exploitation and exploration plus probabilistic (*EE + Prob*) sampling scheme (Algorithm 3). The scheme consists of two sampling stages: a burn-in stage and an adaptive stage. The burn-in stage aims to explore the entire observation space and ensure every observation is sampled several times. During the next adaptive stage, observations with levels of uncertainty greater than a threshold are classified into of observations using probabilistic sampling. Here, $\{\gamma^{(t)}\} \in [0.5,1], t = 1,2,..$ is a monotonically increasing sequence that controls sampling size in the exploitation and exploration step. Meanwhile, the algorithm explores the rest of the observations with uniform weights to avoid exclusively focusing on difficult observations. The reason why we randomly sample the observations that we are confident about is that, we need to include a fair amount of easy-to-cluster observations to construct well-defined clusters in

```
Algorithm 2: Weight updating in adaptive observation sampling scheme
```
the high uncertainty set, and the algorithm exploits this set by sampling $\gamma^{(t)}$ proportion each minipatch so as to better cluster the uncertain ones. We also propose to use the *EE + Prob* scheme as our adaptive feature sampling scheme, which is discussed in Interpretable Minipatch Adaptive Consensus Clustering (IMPACC) section.

**Input:** $S^{(t-1)}, w_I^{(t-1)}, \{I_l\}_{l=1}^{t-1}, \alpha_I; S^{(0)} = 0, w_I^{(0)} = \frac{1}{N}$;

1. Calculate sample uncertainty $u_i = \frac{1}{N}\sum_{i'=1}^{N} S(i, i')(1 - S(i, i')) \times \frac{t-1}{\sum_{l=1}^{t-1} I(i \in I_l)}$;

2. Update observation weight vector $w_I^{(t)} = \alpha_I w_I^{(t-1)} + (1 - \alpha_I)\frac{u}{\sum_{i=1}^{N} u_i}$; **Output:** $w_I^{(t)}$.

In Algorithm 3, $t$ denotes the current count of iterations, $E$ denotes number of burn-in epochs with default value 3, and $w_I^{(t-1)}$ is generated from Algorithm 2. And $\{\tau\}$ is the data-

driven threshold of uncertain observations (important features), which is set to be the 90% quantile of observation weights (mean plus one standard deviation of feature weights).

## Relation to existing literature

Several have suggested similar weight updating approaches in the consensus clustering literature. Ren et al. [22] also obtain observation weights by confusion values as in our method. The difference is that, their method only uses the weighting scheme at the final clustering step rather than adaptive sampling. On the other hand, similar to our adaptive weight updating scheme, Duarte et al. [19], Topchy et al. [21] and Parvin et al. [20] iteratively update weights depending on clustering history. However, these existing methods utilize probabilistic sampling, so they would largely suffer from biased sampling and inaccurate results by only focusing on hard observations. However, instead of probabilistic sampling, we design the *EE + Prob* sampling scheme to leverage the weights, which is inspired by the exploration and exploitation (*EE*) scheme from multi-arm bandits [46,47] and also employed for feature selection with minipatches [36]. Compared to the latter, the innovation in our approach is to combine the advantages of probabilistic sampling and exploitation-exploration sampling, which proves to have particular advantages for clustering. Comparisons with other possible sampling schemes proposed in the literature are in S1 Text.

```
Algorithm 3: Adaptive Observation (Features) Sampling Scheme—EE +Prob
Input:t, n, N, E, {γ(t)}, wᵢ^(t-1), {τ}; wᵢ^(0) = 1/N;
Initialization: Q = [N/n], I = {1,...,N}; if t ≤ E · Q then
// Burn-in stage if mod_Q(t) = 1 then
// New epoch
Randomly reshuffle feature index set I and partition into disjoint
sets {I_Q}_{q=0}^{Q-1};
else Set It = I_modQ(t); end
// Adaptive stage
1. Update observation weights wᵢ^(t) by Algorithm 2;
2. Create high uncertainty set Hᵢ = {i ∈ {1,...,N}: wᵢ^(t) > τ_{wᵢ}^(t)};
3. Exploitation: sample min(n,γ^(t)|Hᵢ|) observations I_{t,1} ⊆ Hᵢ with
probability wIHtI;
4. Exploration: sample (n - min(n,γ^(t)|H|ᵢ)) observations I_{t,2} ⊆ {1,...,
N}\Hᵢ uniformly at random;
5. Set It = It,1 ∪ It,2;
end Output: Iₜ.
```

## Interpretable Minipatch Adaptive Consensus Clustering (IMPACC)

One major drawback of consensus clustering is that it lacks interpretability into important features. This is especially important for high-dimensional data like that in bioinformatics, where we expect only a small subset of features to be relevant for determining clusters. To address this, we develop a novel adaptive feature sampling approach termed Interpretable Minipatch Adaptive Consensus Clustering (IMPACC) that learns important features for clustering and improves clustering accuracy for high-dimensional data. In clustering, two types of approaches to determine important features have been proposed. One is to obtain a sparse solution by solving an optimization problem [29–31], and another one is to rank features by their influence to results [32–34]. However, in data sets with a large number of observations and features, both kinds of methods suffer from significant computational inefficiency. So the question we are interested in is, can we achieve fast, accurate, and reliable feature selection within the consensus clustering process with minipatches? We address this question by proposing a novel adaptive feature weighting method that measures the feature importance in each minipatch

and then ensembles the results to increase the weights of the important features. Given these adaptive feature weights, we can then utilize our adaptive sampling scheme proposed in Algorithm 3 to sample important features more frequently. Outlined in Algorithm 4, we propose an adaptive feature weighting scheme by testing whether each feature is associated with the estimated cluster labels on that minipatch. To do so, we use a simple ANOVA test in part, because it is computationally fast and only requires one matrix multiplication. Based on the p-values from these tests, we establish an important feature set, $A$, and obtain the importance scores as the frequencies of features being classified into this feature set over iterations. Then the feature sampling weights are dynamically updated with learning rate $\alpha_F$, with a default value 0.5. Therefore, by ensembling feature importance obtained from each iteration, we are able to simultaneously improve clustering accuracy and build model interpretability from resulting feature weights, with minimal sacrifices of computation time. In Algorithm 4, $C^{(t-1)}$ denotes the clustering labels on the $(t-1)$-th minipatch, denotes sets of subsampled features in each minipatch up to iteration $t-1$, denotes the feature support constructed up to iteration $t-2$, and the p-value cutoff $\eta$ has default value 0.05. We also explored alternative measures of the association between features and cluster labels in a minipatch. These include using a non-parametric ANOVA (a Kruskal-Wallis test), which relaxes normality assumptions, and using a multinomial regression of features to predict cluster assignments, which can account for feature correlations. Both of these approaches, however, have a higher computational burden than using a simple ANOVA test. We explore these empirically to additionally show that they also yield lower clustering accuracy in S1 Text.

We propose to utilize the same type of *EE + Prob* sampling scheme (Algorithm 3) given our feature weights to learn the important features for clustering. Such a scheme exploits the important features and samples these more frequently as the algorithm progress. But it also balances exploring other features to ensure that potentially important features are not missed. Our final IMPACC algorithm then utilizes both adaptive observation sampling and adaptive feature sampling to improve computation efficiency and clustering accuracy while also providing feature interpretability. Utilizing minipatches in consensus clustering allows us to develop these innovative adaptive sampling schemes and be the first to propose feature learning in this context. Even though IMPACC has several hyper-parameters, in practice, our methods are quite robust and reliable to parameter selections and generally give a strong performance under default parameter settings. Therefore, we are freed from the computationally expensive hyper-parameter tuning process and its computational burdens. We include a study on learning accuracy with different hyper-parameters and default values and suggest a data-driven tuning process in S1 Text. Overall, the proposed MPACC with only adaptive sampling on observation is more suitable for data of no or little sparsity; and IMPACC, which adaptively subsamples both observations and features in minipatch learning, can be more useful when dealing with high dimensional and sparse data sets in bioinformatics. It enhances model accuracy, scalability, and interpretability by focusing on uncertain observations and important features in an efficient manner. Our empirical study in Results section demonstrates the major advantages of the IMPACC method in terms of clustering quality, feature selection accuracy, and computational savings.

**Algorithm 4:** Weight updating in adaptive feature sampling scheme
**Input:**, $w_F^{(t-1)}$, $\alpha_F$; $w_F^{(0)} = 1/M$;
1. For each feature $j \in F_{t-1}$, conduct ANOVA test between features $j$ and $C^{(t-1)}$, record p-value $p_j^{(t-1)}$;
2. Create a feature support $A^{(t-1)} \subseteq F_{t-1} : A^{(t-1)} = \{j \in \{1, \ldots, m\} : p_j^{(t-1)} > \eta\};$ ;
3. Update feature weight vector $w_F^{\ t} \in R^M$ by ensembling feature supports

$\{A^l\}_{l=1}^{(t-1)}$:

$$w_{F_j}^{(t)} = \alpha_F w_{F_j}^{(t-1)} + (1 - \alpha_I) \frac{\sum_{l=1}^{t-1} I(j \in F_j, j \in A^l)}{\max(1, \sum_{l=1}^{t-1} I(j \in F_l))}$$

**Output:** $w_F^{(t)}$.

## Results

In this section, we assess the performance of IMPACC and MPCC with application to a high dimensional and high noise synthetic simulation study in Synthetic Data section and four large-scale real data sets in Case Studies on Real Data section, in comparison with several conventional clustering strategies.

### Synthetic data

We evaluate the performance of MPCC and IMPACC in terms of clustering accuracy and computation time with widely used competitors, and compare IMPACC's feature selection accuracy with the existing sparse feature selection techniques. We propose two kinds of generative models for the synthetic data, with different structures of feature correlation. Here we only show the results of sparse simulation with autoregressive covariance structure. In addition, we also generate synthetic data based on a real single-cell RNA-seq data set using the splatter single-cell simulation method [48]. The results of splatter simulation and simulation with block-diagonal covariance structure conducted in sparse, weak sparse and no sparse scenarios are in S1 Text. In the sparse autoregressive simulation study, each data set is created from a mixture of Gaussian with $AR(1)$ covariance structure, where the covariance between feature $j$ and $j'$ can be written as $\sigma_{j,j'} = \rho^{|j-j'|}$. The parameter $\rho$ is set to be 0.5. We set the number of observations, features and clusters to be $N = 500$, $M = 5,000$, $K = 4$, respectively. In order to better reflect the structure of real bioinformatics data, we design unbalanced cluster sizes and the numbers of observations in each cluster are 20, 80, 120, 280. The means of features in synthetic data is $\mu = [\mu_k, \mu_0]$, where $\mu_k \in \mathrm{R}^{25}$ and $\mu_0 = \mathbf{0}_{4975}$ are the means of 25 signal features and 4,975 noise features, respectively. The signal-to-noise (SNR) ratio is defined as the L2-norm of feature means: $SNR = \|\mu\|_2$. In order to assess feature selection capability, synthetic data is generated with $SNR$ ranging from 1 to 8. Specifically, the signal features are generated with $\mu_1 = \frac{SNR}{5} \cdot 1_{25}, \mu_2 = \left(\frac{SNR}{5} \cdot 1_{13}^T, -\frac{SNR}{5} \cdot 1_{12}^T\right)^T, \mu_3 = \left(-\frac{SNR}{5} \cdot 1_{13}^T, \frac{SNR}{5} \cdot 1_{12}^T\right)^T, \mu_4 = -\frac{SNR}{5} \cdot 1_{25}$. Data with higher SNR ratio has more informative signal features so is easier to be clustered. For all clustering algorithms, we assume oracle number of clusters $K$. Hierarchical clustering is applied as the final algorithm in IMPACC and MPCC, with the number of iterations determined by an early stopping criteria, as described in S1 Text. And we have exactly the same setting as those of MPCC in regular consensus clustering, including the number of iterations. Ward's minimum variance method with Manhattan distance is used in all hierarchical clustering related methods. Details on the implementation of competing methods are in S1 Text. In terms of feature selection, IMPACC provides feature importance scores ranging in [0,1], and sparseKM and sparseHC generate sparse feature weights with zero values for unimportant features. We propose two methods to evaluate feature selection accuracy. The oracle method selects the top 25 features (the oracle number of signal features) with the highest importance score in IMPACC or the highest non-zero weights in sparseKM and sparseHC. And the data-driven feature selection is to select features with importance scores higher than the mean plus one standard deviation of all scores in IMPACC, and select all the features with non-zero weights in sparseKM and sparseHC [30].

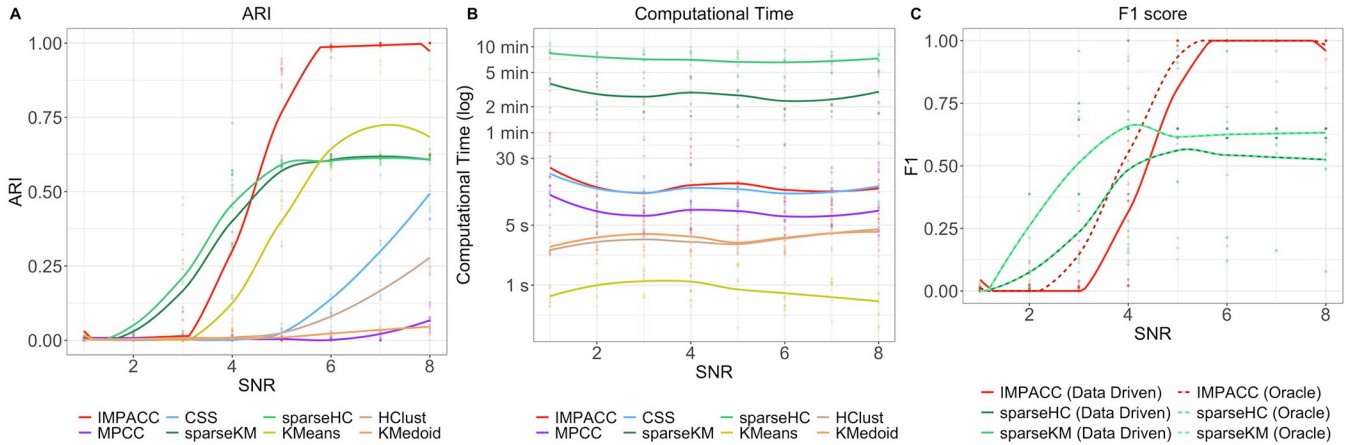

**Fig 1. Clustering performance (ARI), feature selection accuracy (F1 score), and computation time on sparse synthetic data sets.** (A) ARI (higher is better) of estimated grouping; (B) computation time in log seconds; (C) F1 score for signal feature estimates with oracle and data driven selection. IMPACC has superior performance over competing methods in clustering and feature selection accuracy with significant computational savings.

We use adjusted rand index (ARI) to evaluate the clustering performance and the F1 score to measure feature selection accuracy, which both range in [0,1], with a higher value indicating higher accuracy. The averaged results over 10 repetitions are shown in Fig 1. Overall, IMPACC yields the best clustering performance over all competing methods with the highest ARI in most of the *SNR* settings. Comparing feature selection performance, IMPACC has perfect recovery on informative features, with an F1 score equaling to 1 when *SNR* is large, and is significantly better than sparseKM and sparseHC. Note that the oracle and data-driven F1 scores are the same for sparseKM and sparseHC because these two methods under-select important features. Additionally, IMPACC achieves significantly major computational advantages comparing to sparse feature selection clustering strategies. All of the computation time is recorded on a laptop with 16GB of RAM (2133 MHz) and a dual-core processor (3.1 GHz). Note that we only show results of the sparse simulation with autoregressive covariance structure in Fig 1, and we include the rest scenarios in S1 Text. Our methods are still dominant in sparse simulations with block-diagonal structure and splatter simulations, but IMAPCC shows little improvement in the no-sparsity scenario when all the features are relevant.

## Case studies on real data

We apply our methods to one bulk-cell RNA-seq data set, which measures the expression of different tumor cells, three gold-standard single-cell RNA-seq data sets and one image data set

**Table 1. Data sets used in empirical study.**

|                | PANCAN            | Biase        | Goolam       | Yan           | COIL20 |
|----------------|-------------------|--------------|--------------|---------------|--------|
| **Data type**  | RNA-seq           | scRNA-seq    | scRNA-seq    | scRNA-seq     | Image  |
| **Tissue**     | tumor cells       | mouse embryos| mouse embryos| human embryos |        |
| **# clusters** | 5                 | 3            | 5            | 7             | 20     |
| **# observations** | 761           | 49           | 124          | 90            | 1,440  |
| **# features** | 13,244            | 25,737       | 41,480       | 20,286        | 1,024  |
| **% zeros**    | 14.2%             | 50.43%       | 68.56%       | 38.08%        | 34.38% |
| **citation**   | [49]              | [51]         | [52]         | [53]          | [54]   |
| **Source**     | Synapse:syn4301332| GSE57249     | E-MTAB-3321  | GSE36552      |        |

with known cluster labels, whose information is reported in Table 1. The PANCAN bulk RNA-seq data [49] is a benchmark data obtained from *UCI Machine Learning Repository* [50], which contains gene expressions of patients with different types of tumor: BRCA, KIRC, COAD, LUAD and PRAD. The cluster information of the three single-cell RNA-seq data is known because these data sets are generated from cells of various development stages. The Biase [51] data is generated from 49 single cells composed of 1-cell (zygote), mid-stage 2-cell, and 4-cell mouse embryos. The Goolam [52] data investigates gene expression patterns in the pre-implantation development of mouse embryos, including cells isolated from the 2-cell stage to the 32-cell stage. And the Yan [53] data measures gene expression of cells from human pre-implantation embryos and human embryonic stem cells at different passages. In the RNA-seq data, gene expressions are transformed by $x \rightarrow \log_2(1 + x)$ before conducting clustering algorithms; the image data set [54] is adjusted to be within the range [0,1]. Note that we do not conduct any prior feature selection before applying clustering algorithms. With the same settings in Synthetic Data section, we evaluate the learning performance of MPCC and IMPACC with existing methods, with the number of clusters being oracle. Details on the implementation of competing methods are in S1 Text.

Table 2 summarizes the mean of 10 realizations of clustering results on real data sets. IMPACC is either the best or among the top-performing methods in each data set at discovering known clusters with the high ARI scores. Also, it demonstrates major computational advantages, sometimes even beating hierarchical clustering. Clustering followed by dimension reduction via tSNE can have faster and better clustering accuracy for some of the data sets, but it fails to provide direct interpretability of feature importance. Many conduct inference for differentially expressed genes post clustering, but this suffers from selection bias and inflated false positives [55–57]; thus, a direct way to assess important genes as with our method is preferred.

**Table 2. Clustering performance (ARI) and computation time in seconds on real data sets with known cluster labels.**

| | ARI | | | | | Time (s) | | | | |
|---|---|---|---|---|---|---|---|---|---|---|
| | PANCAN | Biase | Goolam | Yan | COIL20 | PANCAN | Biase | Goolam | Yan | COIL20 |
| IMPACC (HC) | **0.991** | 0.953 | 0.815 | 0.742 | 0.74 | 33.379 | 1.849 | 7.602 | 4.17 | 70.297 |
| IMPACC (Spec) | 0.99 | 0.953 | **0.829** | 0.742 | 0.663 | 33.379 | 1.849 | 7.602 | 4.17 | 70.297 |
| MPCC (HC) | 0.982 | 0.948 | 0.66 | 0.742 | 0.717 | 21.897 | 0.093 | 4.73 | 2.189 | 52.446 |
| MPCC (Spec) | 0.991 | 0.948 | 0.682 | 0.772 | 0.672 | 21.897 | 0.093 | 4.73 | 2.189 | 52.446 |
| Consensus (HC) | 0.754 | 0.953 | 0.452 | **0.834** | 0.673 | 75.377 | 0.121 | 1.797 | 0.967 | 557.623 |
| Consensus (Spec) | 0.774 | 0.953 | 0.684 | 0.763 | 0.67 | 75.377 | 0.121 | 1.797 | 0.967 | 557.623 |
| sparseKM | 0.981 | **1** | 0.459 | 0.736 | 0.441 | 1044.875 | 46.636 | 141.162 | 68.011 | 95.572 |
| sparsHC | | 0.342 | 0.514 | 0.777 | | | 14.883 | 86.904 | 22.97 | |
| Seurat | | 0.66 | 0.447 | 0.548 | | | 0.922 | 1.412 | 0.791 | |
| SC3 | | 0.948 | 0.687 | 0.731 | | | 75.290 | 73.234 | 68.598 | |
| tSNE+KMeans | 0.983 | 0.509 | 0.317 | 0.736 | 0.619 | 7.853 | 0.288 | 1.598 | 0.514 | 133.09 |
| tSNE+HC | 0.991 | 0.948 | 0.3 | 0.671 | 0.685 | 7.864 | 0.287 | 1.598 | 0.514 | 3.543 |
| tSNE+spectral | 0.803 | 0.948 | 0.307 | 0.666 | **0.787** | 12.598 | 0.411 | 1.685 | 0.594 | 3.578 |
| tSNE+KMedoid | 0.98 | 0.948 | 0.354 | 0.641 | 0.727 | 8.008 | 0.287 | 1.6 | 0.516 | 20.255 |
| KMeans | 0.795 | 0.948 | 0.493 | 0.544 | 0.771 | 2.67 | 0.045 | 0.326 | 0.107 | 7.04 |
| HClust | 0.756 | 0.948 | 0.433 | 0.763 | 0.54 | 57.236 | 0.057 | 1.116 | 0.201 | 0.201 |
| Spectral | 0.734 | 0.948 | 0.381 | 0.473 | 0.65 | 4.817 | 0.118 | 0.393 | 0.181 | 2.097 |
| KMedoid | 0.761 | **1** | 0.676 | 0.743 | 0.447 | 58.955 | 0.066 | 1.212 | 0.214 | 12.491 |

Clustering performance (ARI) and computation time in seconds on real data sets with known cluster labels. The IMPACC method is among the best in terms of clustering performance, with significant improvements on the computational cost compared to sparseKM, sparseHC, and consensus clustering. The MPCC method also yields comparable clustering performance and computational speed.

Even though single cell RNA-seq specific method SC3 has comparable accuracy in the Biase [51] and Yan [53] data set, these methods select genes with high variance before performing clustering algorithm and do not provide inherent interpretations of important genes. Note that R failed to apply sparseHC to large genomics data due to excessive demand on computing memory, and we only perform SC3 and Seurat on single-cell RNA-seq data sets. Further, even though MPCC has a slightly lower ARI than IMPACC, it still yields better or comparable performance in learning accuracy over consensus and standard methods, and it is relatively fast. Additionally, we visualize the consensus matrices of IMPACC and compare them to that of regular consensus clustering in Fig 2. We can conclude that IMPACC is able to produce more accurate consensus matrices, with clearer diagonal blocks of clusters and less noise on off-diagonal entries.

## Interpretability analysis on Yan data set

IMPACC further provides interpretability in terms of feature importance. Since the feature support set in IMPACC is constructed by including features that demonstrate different expressions across clusters, we can identify differentially expressed genes from the final feature importance scores. We propose a data-driven way to set the cutoff as the mean plus one standard deviation of all scores to conduct feature selection. Here we conduct our interpretability analysis focusing on a realization of IMPACC clustering on the Yan [53] data set. IMPACC selects 466 differentially expressed genes using the data-driven cutoff, and the full list of genes is reported in S1 Table. We plot the gene expression matrix of the top 50 differentially expressed genes determined by IMAPCC in Fig 3, with the subgroups defined by the final consensus matrix of IMPACC, using the oracle number of clusters $K = 7$ (separated by white vertical lines). The important genes selected by IMPACC have significantly different expressions among different clusters of cells, especially in the Morulae cluster.

The Yan [53] data set measures gene expression in human oocytes, early embryos at seven developmental stages and hESC cells. And the original paper Yan et al. [53] discovered that the EPI cells have lower gene expression in gamete generation, germ cell development and reproduction process, indicated from the GO terms identified by differential genes between EPI

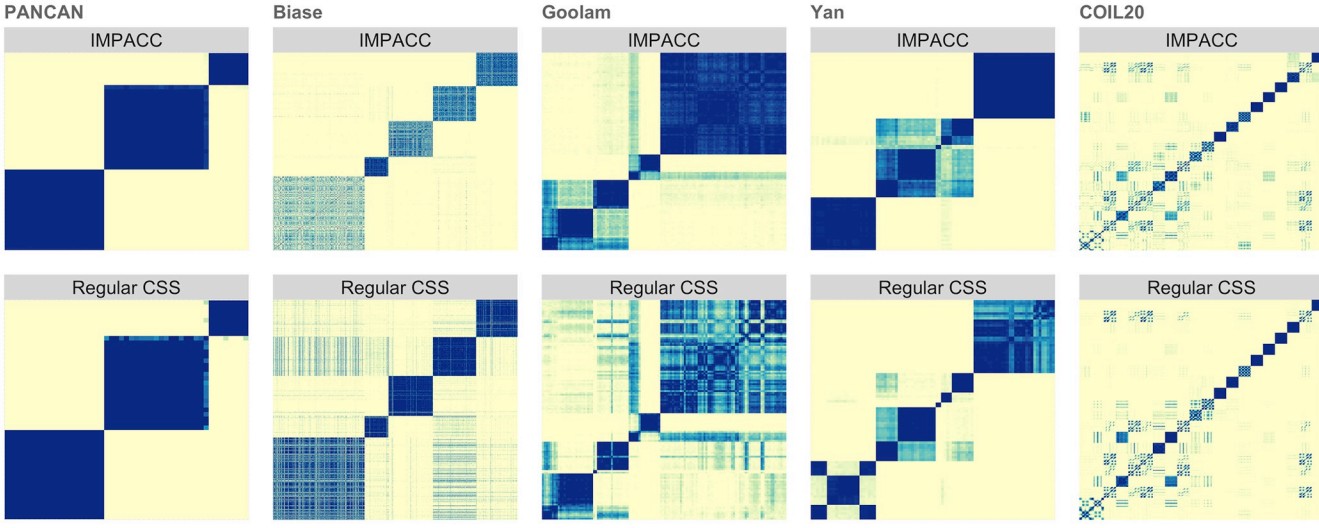

**Fig 2. Heatmaps of final consensus matrix derived from IMPACC and consensus clustering respectively, using oracle number of clusters.** Darker color indicates higher consensus value.

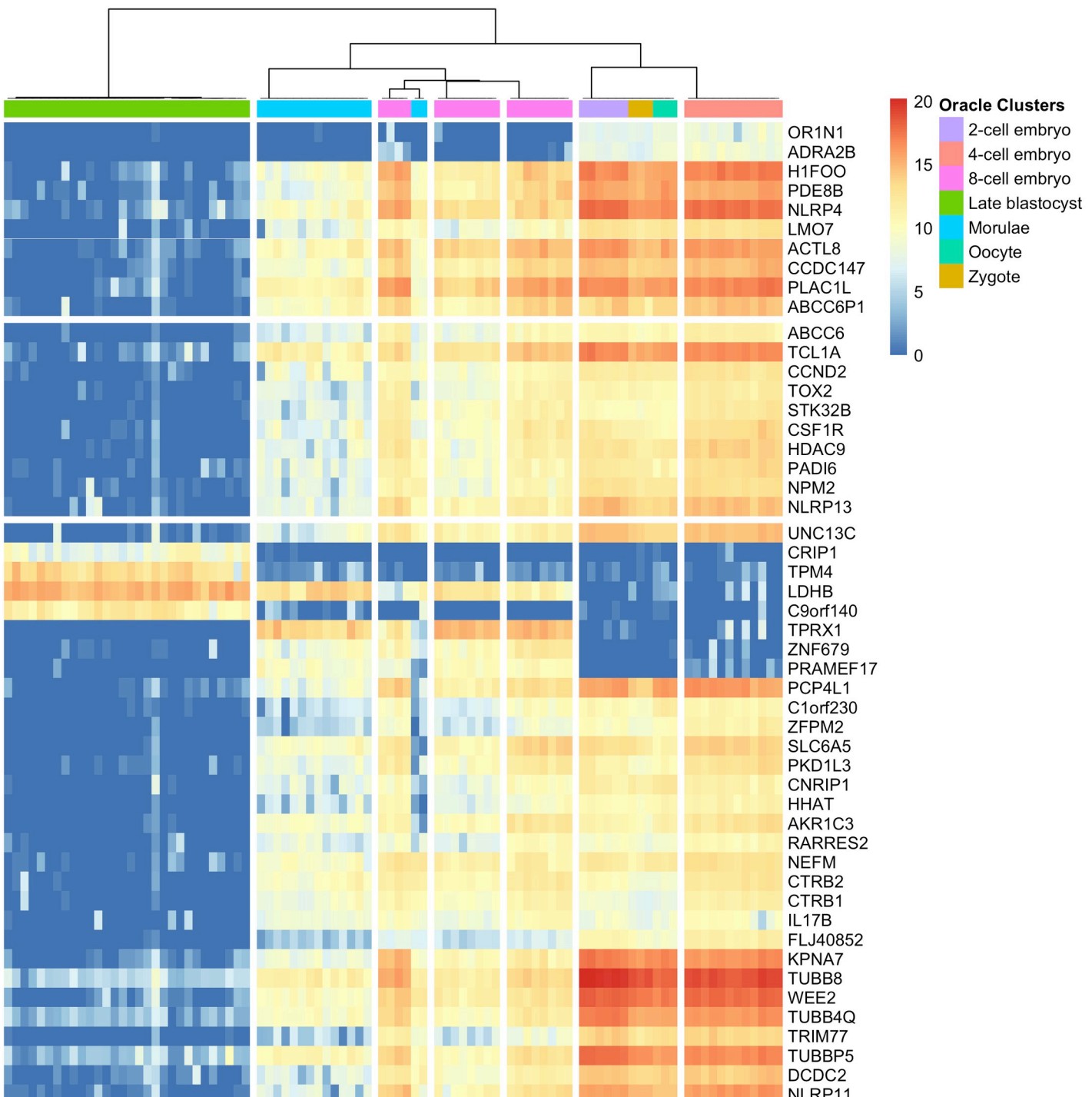

**Fig 3. Gene expression matrix of the top 50 differentially expressed genes identified by IMPACC in Yan data set, with subgroups defined by the final consensus matrix of IMPACC.**

cells and other cell lineages in blastocysts. To further evaluate the model interpretability of IMPACC, we perform Gene Ontology (GO) pathway enrichment analysis on the 466 differentially expressed genes determined by our data-driven approach. With a p-value cutoff of 0.05,

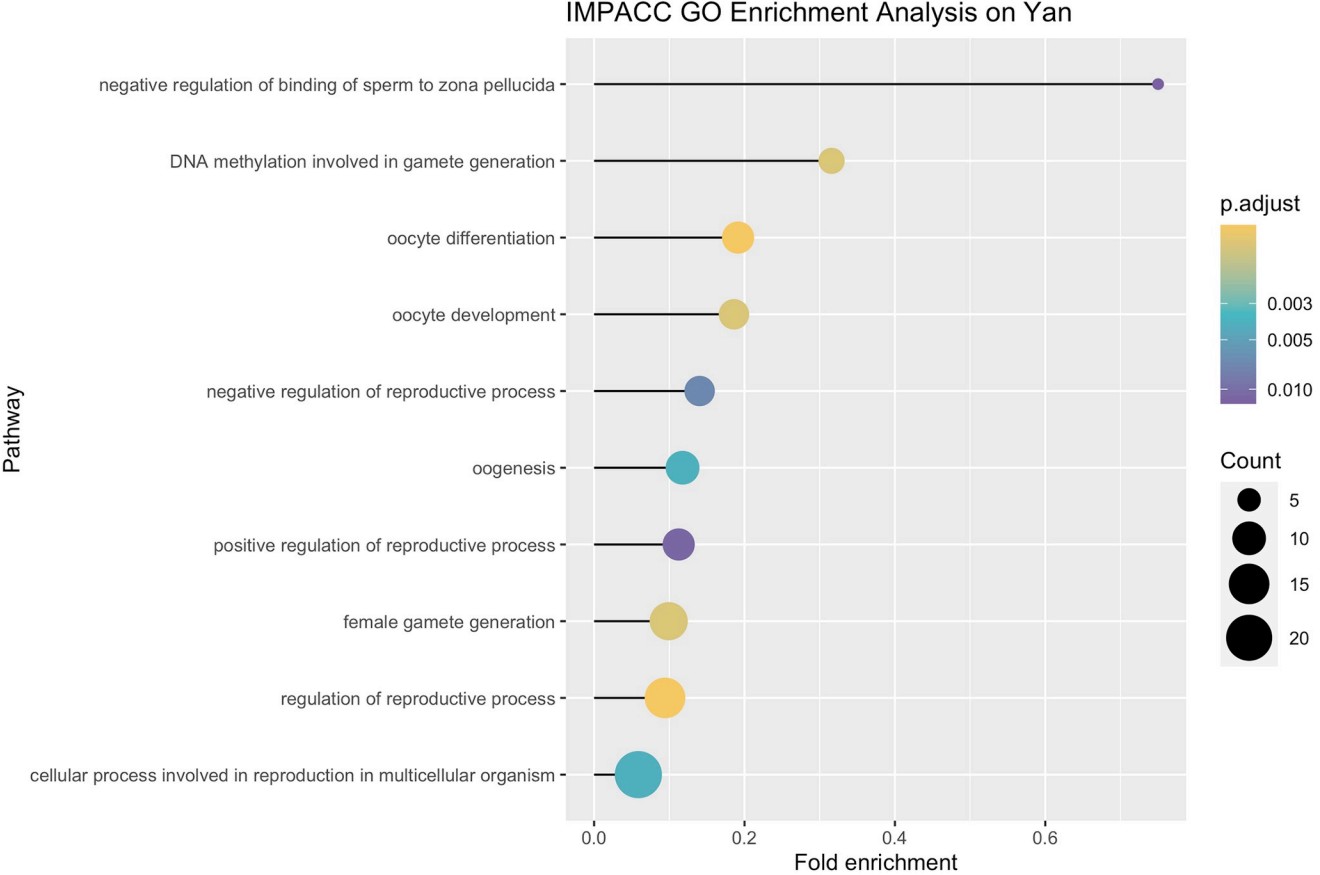

**Fig 4. Top 10 pathway of GO enrichment analysis using the differentially expressed genes identified by IMPACC in Yan data set, with information on adjusted p-values, fold enrichment and count.**

we can successfully identify 26 GO terms, and these pathways are highly related to germ cell development (oogenesis, oocyte development, oocyte differentiation), gamete generation (female gamete generation, DNA methylation involved in gamete generation), and reproduction (regulation of reproductive process, negative regulation of reproductive process, positive regulation of reproductive process, cellular process involved in reproduction in multicellular organism). The top 10 pathways with fold enrichment, p-values, and counts are illustrated in Fig 4. And the enriched GO terms reported in Yan et al. [53], the complete list of GO terms based on IMPACC, and pathway analysis using differentially expressed genes identified by SC3 and sparseKM are in S1 Text and S2 Table. Overall, these results reveal that IMPACC is able to provide accurate and reliable interpretations of scientifically important genes as well as biologically meaningful GO enrichment analysis; these results match the original paper's scientific conclusions in which the cell types are known.

## Discussion

We have proposed novel and powerful methodologies for consensus clustering using minipatch learning with random or adaptive sampling schemes. We have demonstrated that both MPCC and IMPACC are stable, robust, and offer superior performance than competing methods in terms of accuracy. Further, our approaches offer significant computational savings with

runtime comparable to hierarchical or spectral clustering. Finally, IMPACC offers interpretable results by discovering features that differentiate clusters. This method is particularly applicable to sparse, high-dimensional data sets common in bioinformatics. Our empirical results suggest that our method might prove particularly important for discovering cell types from single-cell RNA sequencing data. Note that while our methods offer computational advantages over consensus clustering for all settings, our method does not seem to offer any dramatic improvement in clustering accuracy for non-sparse and non-high-dimensional data sets. In future work, one can further optimize computations through memory-efficient management of the large consensus matrix and through hashing or other approximate schemes. Overall, we expect IMPACC to become a critical instrument for clustering analyses of complicated and massive data sets in bioinformatics as well as a variety of other fields.

## Supporting information

**S1 Text. Fast and Interpretable Consensus Clustering via Minipatch Learning: Supplementary Materials.**
(DOCX)

**S1 Table. Differentially expressed genes in Yan selected by IMPACC.**
(CSV)

**S2 Table. SC3's GO Enrichment Pathway Analysis of Yan data set.**
(CSV)

## Acknowledgments

The authors would like to thank Zhandong Liu and Ying-Wooi Wan for helpful discussions on single-cell sequencing as well as Tianyi Yao for helpful discussions on minipatch learning.

## Author Contributions

**Conceptualization:** Genevera I. Allen.

**Data curation:** Luqin Gan.

**Formal analysis:** Luqin Gan.

**Funding acquisition:** Genevera I. Allen.

**Investigation:** Luqin Gan.

**Methodology:** Luqin Gan.

**Project administration:** Genevera I. Allen.

**Software:** Luqin Gan.

**Supervision:** Genevera I. Allen.

**Validation:** Luqin Gan.

**Visualization:** Luqin Gan.

**Writing – original draft:** Luqin Gan, Genevera I. Allen.

**Writing – review & editing:** Luqin Gan, Genevera I. Allen.

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
