## [Decision Letter · Decision Letter 0]

15 Mar 2022

Dear Ms. Gan,

Thank you very much for submitting your manuscript "Fast and Interpretable Consensus Clustering via Minipatch Learning" for consideration at PLOS Computational Biology.

As with all papers reviewed by the journal, your manuscript was reviewed by members of the editorial board and by several independent reviewers. In light of the reviews (below this email), we would like to invite the resubmission of a significantly-revised version that takes into account the reviewers' comments.

We cannot make any decision about publication until we have seen the revised manuscript and your response to the reviewers' comments. Your revised manuscript is also likely to be sent to reviewers for further evaluation.

Sincerely,

Isidore Rigoutsos, Ph.D.

Associate Editor

PLOS Computational Biology

Mark Alber

Deputy Editor

PLOS Computational Biology

Reviewer's Responses to Questions

**Comments to the Authors:**

Reviewer #1: The work by Gan and Allen describes the development of an interesting algorithm that can solve the limitations of consensus clustering when applied to large-scale biological data. The manuscript is well written, organized and largely seems technically sound. However, I have some questions for the authors.

1. Throughout the manuscript, authors claim that in high-dimensional biological data, we expect only a small subset of features to be relevant for determining clusters. Although this can be true in some cases, I can think of many scenarios where this is not true (even in some of the datasets they analyze). For example, in a biopsy of a tumor within a normal tissue, the tumor cells have a vastly different gene expression profile. Even within a normal tissue, let's say bone marrow, we expect to find a wide variety of cell types, each one with many distinct genes expressed and common ones expressed at different levels. This is not to say that their method is not applicable to such datasets but I would expect that the authors consider and discuss the implications in such cases.

2. In at least two instances (lines 141 and 248), authors choose a relatively simple methodology so that the overall algorithm is fast enough. To test that this is the best choice (not in terms of speed but in terms of accuracy), I would expect to see a side-by-side comparison with additional tests/methods and show that speed is not compromising accuracy.

3. Lines 131-151 deal with the problem of accuracy. I am not sure that I understand how this approach specifically addresses the problem of accuracy. By the end of this paragraph, I can understand the methodology but not how it achieves better accuracy (only how it proposes to do so).

4. The biological interpretations are extremely weak (lines 342-355). Authors perform KEGG pathway analysis (not described with which tool, statistical thresholds?) and argue that some pathways are important in one setting vs. the other. This is a subjective analysis and I can argue for the opposite in some cases: Why is insulin secretion important in IMAPCC of Table 3 when we are focusing on brain cells? Why is this not a false positive? Authors need to look at the genes within each pathway in more detail. They also need to justify their findings biologically and compare them to the ones reported in the original papers that generated the data they use. In addition, they need to perform some basic correlation analysis of importance weights, pathway fold enrichments (not just p values) among the three tested approaches.

Reviewer #2: Review uploaded as attachment

Reviewer #3: In this manuscript, the authors described a study that combined Minipatch learning and adaptive sampling to improve the current consensus clustering method. The authors showed that the new consensus clustering methods could achieve higher computational efficiency, improved accuracy and better interpretability, on both synthetic and real-world data. In general, the manuscript is very well written and easy to follow. The statistical modeling and validation approaches are sound to me. As consensus clustering is widely used in biological data analysis, a more robust and interpretable clustering method could serve as a very useful tool for the research community. Therefore, I recommend the publication of this work after resolving some minor issues.

1. The authors provided the codes for the implementation of the described methods. However, the documentation for the usage of those codes is lacking, which may make it difficult for other people to use this tool. The authors are encouraged to organize those individual R scripts into an R package (ideally a Bioconductor package) with proper documentation (function help page, tutorial, example and so on.)

2. The authors used way too many “dramatic” in this manuscript. In my opinion, it’s better to refrain from using such words, especially when the improvement of computational efficiency and accuracy in some scenarios is rather comparable to some other clustering methods and the performance also depends on whether the dataset itself is sparse or not, as mentioned by the authors in the discussion part.

3. The authors are encouraged to strengthen the hyper-parameter tuning part and better justify the generalizability of the recommended hyper-parameters. The authors showed that hyper-parameters had a limited impact on model performance and therefore suggested that users could choose the default parameters. However, the authors only tuned and compared the hyper-parameters on two real-world biological datasets (Brain cells and PANCAN). As the users will perhaps have much more diverse biological datasets, the authors may want to test the hyper-parameters in more cases (other types of biological data or synthetic data) to ensure the generalizability of the recommended hyper-parameters

**Have the authors made all data and (if applicable) computational code underlying the findings in their manuscript fully available?**

Reviewer #1: Yes

Reviewer #2: Yes

Reviewer #3: Yes

PLOS authors have the option to publish the peer review history of their article (what does this mean?). If published, this will include your full peer review and any attached files.

Reviewer #1: No

Reviewer #2: No

Reviewer #3: No
---

## [Decision Letter · Decision Letter 1]

25 Jun 2022

Dear Ms. Gan,

Thank you very much for submitting your manuscript "Fast and Interpretable Consensus Clustering via Minipatch Learning" for consideration at PLOS Computational Biology. As with all papers reviewed by the journal, your manuscript was reviewed by members of the editorial board and by several independent reviewers. The reviewers appreciated the attention to an important topic. Based on the reviews, we are likely to accept this manuscript for publication, providing that you modify the manuscript according to the review recommendations.

Sincerely,

Isidore Rigoutsos, Ph.D.

Associate Editor

PLOS Computational Biology

Mark Alber

Deputy Editor

PLOS Computational Biology

[LINK]

Reviewer's Responses to Questions

**Comments to the Authors:**

Reviewer #1: Authors have adequately addressed most of my comments. However, I still have a minor comment on the interpretation of the biological results. Figure 3 of the revised manuscript shows the differentially expressed genes that they identify through their methods and figure 4 shows the relevant pathways. These mainly include genes/pathways down-regulated as development progresses. However, in the main manuscript they argue that they "successfully identify 26 GO terms, and these pathways are highly related to the regulation of the reproductive process and cell development" (page 15; line 418). In addition they mention that in the original publication Yan et al. "discovered that the differential genes between EPI cells and the remaining cells are enriched for GO terms related to transcriptional regulation and germ cell development". Why is germ cell development (e.g. oogenesis and oocyte differentiation) relevant when the biological context is pre-implantation development? I think the authors need to better clarify the parts of biology that their methodology captures. One way of doing so would be to better argue on the importance of these pathways in this context (it makes sense biologically for the negative regulators of sperm binding to the zona pellucida to be downregulated after proper fertilization) but I would expect a much more thorough discussion. Another way is to show some Venn diagrams of the identified pathways with pathways relevant in mouse pre-implantation development. This is a well-studied period of development with many datasets and results publicly available that the authors can utilize and justify the robustness of their results or to better showcase their own differential expression analysis.

Reviewer #2: The authors provide a much improved revision, with appropriately moderated claims, which largely addresses the original review criticisms. Residual comments below are minor and should be easy to address (some are just suggestions).

COMMENTS

Several notation inconsistencies still remain in the pseudocode. The authors should go over everything CAREFULLY and fix. E.g., in algorithm 1, Cit should rear Ci^(t)^.

For all algorithms, initial values of variables should be specified. E.g., in Algorithm 2, specify S(t=0) and wI(t=0).

“Here we only show the results of sparse simulation with autoregressive covariance structure, as it is the best representative of high dimensional bioinformatics data.” Justification for this statement (“best representative of high dimensional bioinformatics data”) should be provided, e.g., via citing work where this is demonstrated. And a rationale for the specific choice of covariance (σj,j′ = ρ|j−j′|) should be given.

In my opinion, the results of clustering the Splatter-simulated data are more reflective of the algorithm’s performance on real scRNA-seq data and, in fairness, should at the very least be presented in the main text along with the results from the autoregressive model, rather than in the appendix.

One recommendation for Table 2: in addition to displaying the actual measurements, consider using color gradients for the table cells (similar to how you present data in Appendix Figure 21), to aid visual delineation of “good” and “bad” values.

“Clustering followed by dimension reduction via tSNE can have faster and better clustering accuracy for some of the data sets, but they fail to provide interpretability in terms of feature importance”. I disagree with the authors that this is a significant limitation. E.g., one can always run a post-clustering ANOVA (or similar) to prioritize features, if desired.

For the PANCAN dataset, table 1 claims that it contains five clusters. Where is this number coming from?

Reviewer #3: The authors have adequately addressed my previous comments.

**Have the authors made all data and (if applicable) computational code underlying the findings in their manuscript fully available?**

Reviewer #1: Yes

Reviewer #2: Yes

Reviewer #3: Yes

PLOS authors have the option to publish the peer review history of their article (what does this mean?). If published, this will include your full peer review and any attached files.

Reviewer #1: No

Reviewer #2: No

Reviewer #3: No

Figure Files:

Data Requirements:

Reproducibility:

References:

---

## [Decision Letter · Decision Letter 2]

15 Sep 2022

Dear Ms. Gan,

We are pleased to inform you that your manuscript 'Fast and Interpretable Consensus Clustering via Minipatch Learning' has been provisionally accepted for publication in PLOS Computational Biology.

Best regards,

Isidore Rigoutsos

Academic Editor

PLOS Computational Biology

Mark Alber

Section Editor

PLOS Computational Biology

Reviewer's Responses to Questions

**Comments to the Authors:**

Reviewer #1: Authors have addressed my concerns

**Have the authors made all data and (if applicable) computational code underlying the findings in their manuscript fully available?**

Reviewer #1: Yes

PLOS authors have the option to publish the peer review history of their article (what does this mean?). If published, this will include your full peer review and any attached files.

Reviewer #1: No

---

## [Editor Report · Acceptance letter]

23 Sep 2022

PCOMPBIOL-D-21-02086R2 

Fast and Interpretable Consensus Clustering via Minipatch Learning

Dear Dr Gan,

I am pleased to inform you that your manuscript has been formally accepted for publication in PLOS Computational Biology. Your manuscript is now with our production department and you will be notified of the publication date in due course.

With kind regards,

Olena Szabo
